The behavior of sympatric sea urchin species across an ecosystem state gradient

http://orcid.org/0000-0002-0141-7495 Belleza Dominic Franco C. 1 dfcbelleza@gmail.com
Urae Takeshi 1
Tanimae Shin-ichiro 1
Toyama Kento 2
Isoda Akari 2
http://orcid.org/0000-0003-2543-1301 Nishihara Gregory N. 1 3
1 Graduate School of Fisheries and Environmental Sciences, Nagasaki University , Nagasaki City, Nagasaki , Japan
2 Faculty of Fisheries, Nagasaki University , Nagasaki City, Nagasaki , Japan
3 Organization for Marine Science and Technology, Institute for East China Sea Research, Nagasaki University , Nagasaki City, Nagasaki , Japan
Waiho Khor
Electronic publication date: 2023 Jun 13
Publication date: 2023
Volume: 11
Electronic Location ID: e15511
Received 2022 Oct 12; Accepted 2023 May 15
Copyright: © 2023 Belleza et al.
Copyright year: 2023
Copyright holder: Belleza et al.
License: This is an open access article distributed under the terms of the Creative Commons Attribution License, which permits unrestricted use, distribution, reproduction and adaptation in any medium and for any purpose provided that it is properly attributed. For attribution, the original author(s), title, publication source (PeerJ) and either DOI or URL of the article must be cited.
License URL: https://creativecommons.org/licenses/by/4.0/

Keywords: Sea urchin behavior, Ecosystem state, Microhabitat, Seaweed ecosystem

Funding: Grant-in-Aid for Scientific Research 17KT0149, 20H03076, and C-#40508321 Japan Society for the Promotion of Science (JSPS) Japanese Ministry of Education, Culture, Sports, Science, and Technology (MEXT) This study was supported by the Grant-in-Aid for Scientific Research: 17KT0149, 20H03076, and C-#40508321 from the Japan Society for the Promotion of Science (JSPS) and the Japanese Ministry of Education, Culture, Sports, Science, and Technology (MEXT). The funders had no role in study design, data collection and analysis, decision to publish, or preparation of the manuscript.

==============================
Background

In temperate macroalgal forests, sea urchins are considered as a keystone species due to their grazing ability. Given their potential to shape benthic communities, we monitored the habitat use by three sympatric sea urchin species and compared their behaviors in a vegetated habitat (VH) and an adjacent isoyake habitat (IH).

Methods

We monitored the environmental conditions and sea urchin density along deep and shallow transects of the VH and IH for over a year. The benthic rugosity at both sites were also surveyed. A mark-recapture experiment was conducted on the two most abundant sea urchins, Diadema setosum and Heliocidaris crassispina, to elucidate sea urchin movement patterns and group dynamics.

Results

We found that exposure to waves was highest at the VH while the IH was sheltered. The deep IH experienced the least amount of light due to high turbidity. Water temperature patterns were similar across sites. The VH benthic topography was more rugose compared to the smoother and silt-covered IH substate. Peak macroalgal bloom occurred three months earlier in IH, but macroalgae persisted longer at the shallow VH. Among the sympatric sea urchins, H. crassispina was most abundant at the shallow VH and was observed in pits and crevices. The most abundant across IH and in the deep VH was D. setosum, preferring either crevices or free-living, depending on hydrodynamic conditions. The least abundant species was D. savignyi, and most often observed in crevices. Small and medium sea urchins were most often observed at the IH site, whereas larger sea urchins were more likely observed at the VH. The mark-recapture study showed that D. setosum was found to displace further at the IH, and H. crassispina was more sedentary. Additionally, D. setosum was always observed in groups, whereas H. crassispina was always solitary.

Discussion

The behaviors of sympatric urchins, Diadema savignyi, D. setosum and H. crassispina, differed in response to changes in the benthic environment and physical conditions. Sea urchin displacement increased when rugosity and wave action were low. Habitat preference shifted to crevices in seasons with high wave action. In general, the mark-recapture experiment showed that sea urchins displaced further at night.

Introduction

The resilience of natural ecosystems is limited and highly dependent on many factors including effect size, persistence, and the synergistic effects of multiple disturbances (Nyström, Folke & Moberg, 2000; Benedetti-Cecchi et al., 2001; Strain et al., 2014). A unique characteristic of an ecosystem is the ability to switch among a variety of alternative states (Scheffer et al., 1993, 2001; Carpenter, Ludwig & Brock, 1999; Xu et al., 2015). Some examples are coral-macroalgal phase shifts (McManus & Polsenberg, 2004), shift from clear to turbid phase in lakes (Scheffer et al., 1993), and forest-open landscape phase shifts (Xu et al., 2015). Macroalgae ecosystems are no exception, because subtidal vegetated ecosystems can collapse into a state where the rocky substrate is bare of macrophytes and habitat complexity and primary productivity are low (Filbee-Dexter & Scheibling, 2014; Krumhansl et al., 2016; Pessarrodona et al., 2021). Such phase shifts (i.e., degradation of macroalgal forests) can be caused by both abiotic (Seymour et al., 1989; Airoldi, 1998; Coleman et al., 2008; Provost et al., 2017) and biotic disturbance (Yamaguchi et al., 2010; Poore et al., 2012; Vergés et al., 2016), and these events have been documented for more than 100 years in Japan, where algae bed degradation is known as isoyake (Fujita, 2010).

Isoyake is a Japanese concept of seaweed deforestation caused by a combination of fluctuating oceanographic conditions and increasing herbivory (Graham, 2010). Compared to sea urchin barrens where sea urchin overgrazing is the main cause, the term isoyake is used to describe the degraded state of a temperate algal bed where the cause is less clearly defined and implies that some phenomenon has caused the deforestation or degradation of algal forests (Graham, 2010; Eger et al., 2022; Sato et al., 2022). In particular, the life cycle of canopy-forming seaweeds in Japan have experienced a marked decrease in growth and biomass in summer, associated with warm surface temperatures and low nutrient supply (Haroun, Yokohama & Aruga, 1989; Serisawa et al., 2004). Unfortunately, oceanographic studies in the East China Sea region show steadily increasing sea surface temperatures of about 0.3 °C year−1 and stratification of the water-column (Son et al., 2012; Lee & Kim, 2013). The rapidly changing environment has created unsuitable conditions for economically important seaweeds to thrive while conventional seaweed production is now more sensitive to climatic and human disturbances (Chen, 2019).

Sea urchins are highly successful benthic grazers because their natural population control is limited to a few specialized predators (Estes & Palmisano, 1974; Fujita et al., 2013; Kawamata et al., 2016). Although they are often considered herbivores, sea urchins have flexible dietary requirements that enable them to switch to omnivory (Agnetta et al., 2013; Rodríguez-Barreras et al., 2015; Leclerc et al., 2021) and cannibalism at higher densities (LeGault & Hunt, 2016) or in rare cases, even directly attack their predators (Clements, Dupont & Jutfelt, 2021). This dietary flexibility enables sea urchins to exploit and overwhelm primary producers in rocky reefs and cause a phase shift (Tuya, Martin & Luque, 2004; Lauzon-Guay, Scheibling & Barbeau, 2009; Flukes, Johnson & Ling, 2012), while maintaining the barren state by subsisting on a mixed diet of small algae and benthic invertebrates (Bonaviri et al., 2011). In pristine algae bed ecosystems, sea urchin populations are controlled by predation (Tegner & Levin, 1983; McKay & Heck, 2008; Gregr et al., 2020). Here, the abundant supply of drift algae provides sea urchins with a reliable food source and they are less likely to forage on the standing algal biomass (Kriegisch et al., 2019).

Sea urchins avoid direct competition with other sea urchin species through omnivory and resource partitioning (Contreras & Castilla, 1987; McClanahan, 1988; Vanderklift, Kendrick & Smit, 2006). Stable isotope analysis suggests that sympatric sea urchin species occupy different trophic levels (i.e., herbivory vs omnivory) despite habitat overlap (Vanderklift, Kendrick & Smit, 2006). For example, specialized structures such as modified aboral podia in Loxechinus albus allows the capture of floating algal pieces while Tetrapygus niger specializes in the efficient consumption of benthic algae due to its longer pyramid structures in the Aristotle’s lantern (Contreras & Castilla, 1987). Benthic communities, depth and temperature can vary widely across latitudes (Guidetti & Dulčić, 2007; Williams, Coleman & Jordan, 2020). This results in large (Leclerc et al., 2021) or few variations (Vanderklift & Wernberg, 2010) in the diet and feeding behavior of conspecific urchins. Nevertheless, this makes sea urchins highly adaptable to their environment and a particular species may display different behavioral patterns when in an algal bed or when in a barren (Yusa & Yamamoto, 1994; Urriago Suarez et al., 2021). However, there is little evidence of simultaneous comparison between both states in field conditions because contrasting states from the same ecosystem are often separated by several decades (Krumhansl et al., 2016).

Our study explores the benthic community of a small embayment where two ecosystem states are temporally co-occurring at a relatively small spatial scale (i.e., 100’s of m). We monitored the abundance and distribution of three sympatric sea urchin species, Diadema savignyi, Diadema setosum and Heliocidaris crassispina, along a vegetated and an isoyake area for over a year (September 2020–December 2021) and also analyzed the movement patterns of the two most abundant sympatric sea urchin species, D. setosum and H. crassispina through a mark-recapture experiment. Morphologically, Diadematid sea urchins (i.e., D. savignyi, D. setosum) differ greatly from H., crassispina. Adult Diadematid sea urchins have larger test sizes (6–9 cm) compared to adult H. crassispina (5.5–7 cm), and also have long and thin venomous spines up to 20 cm, while H. crassispina have short and hard, but non-venomous spines (3–5 cm). Whereas H. crassispina has a dark purple color, both Diadematid species are black, with D. savignyi having distinct bright blue lines along the ambulacral zones while D. setosum have five white spots on the aboral side, one each along the ambulacral zone. Diadematid urchins are known to form aggregations (Pearse & Arch, 1969) while H. crassispina are solitary (Yusa & Yamamoto, 1994). Because of the differences between species, we hypothesized that the substrate component and environmental conditions affect sea urchin behavior and movement patterns while sea urchin species and size determine microhabitat preference.

Materials and Methods

Study site

The study was conducted at Arikawa Bay (Fig. 1), located in northeastern Nakadori Island of the Goto Islands, Nagasaki Prefecture, Japan (129.11°E, 32.99°N). The study site presents a case of contrasting ecosystem states where a vegetated and an isoyake habitat are spatially adjacent but physically separated by a concrete structure (Fig. 1, inset).

Figure 1 Study site map.

Map of the Goto Islands showing the location of Arikawa Bay and the study sites (inset) monitored in the study.

Arikawa Bay is exposed to wind and waves from the north. In general, the coastline is rocky while some parts are armored with concrete walls and wave-breakers. We examined two distinct ecosystems: (A) a vegetated habitat, hereafter known as VH and (B) an isoyake habitat, hereafter known as IH. Within each ecosystem, a 20-m stretch of rocky reef was selected for monitoring in deep (4.27 ± 0.60 m mean ± SD at IH and 4.62 ± 0.62 m at VH) and shallow areas (2.07 ± 0.60 m at IH and 3.06 ± 0.67 m at VH).

The VH is open to the north (i.e., windward side of a concrete jetty) and has a mean wind fetch of 2,342 m (number of vectors reaching maximum limit: 13) (Sato et al., 2022) (see Fig. S1). The substrate is mostly consolidated rocks and small to medium-sized boulders. The VH is dominated by small algae (Corallina spp., Gelidium spp.) and turf algae present year round and sparse patches of perennial Sargassum macrocarpum. Blooms of seasonal macroalgae (i.e., Sargassum horneri, Asparagopsis taxiformis, Dictyopteris spp., Codium fragile, Colpomenia sinuosa and Hydroclathrus clathratus) occur in mid- to late- spring and persist until early to mid-summer. The IH is located behind a concrete jetty and is sheltered from waves approaching from the north (Fig. 1, see inset) and has a mean wind fetch of 1,889 m (12). The IH substrate is also rocky, however silty patches occur in the deeper areas. The rocky surfaces are bare, except for a few patches of encrusting coralline algae apart from the seasonal macroalgal bloom (i.e., S. horneri, C. sinuosa, H. clathratus) that occurs from spring to early summer.

Field monitoring

To determine changes in the benthic environment, we conducted monitoring activities by skin diving in the daytime (ca. 07:00–08:00), along marked sections in the deep and shallow reefs of VH and IH. A 20-m rope was secured to the substrate on both ends and set parallel to shore to guide the installation of a transect tape every monitoring period. Photos of the substrate were taken using a 1 m tall, 1 m2 polyvinyl chloride (PVC) photoquadrat (Preskitt, Vroom & Smith, 2004) with a GoPro Hero 8 camera at 1 m intervals along each belt transect. A total of 10 photos were taken for each belt transect every monitoring activity. For the sea urchin monitoring, all sea urchins observed within a 2-m-wide swath at 1 m intervals (2 × 1 m) along the transect were recorded for their species and whether they occupied pits, crevices, or were free-living (Yusa & Yamamoto, 1994; Gravem & Adams, 2012; Frey & Gagnon, 2016). A total of 10, 2 × 1 m swaths were examined for each belt transect every survey period and the body diameter of all sea urchin species found were measured using a caliper in situ. Sea urchin sizes were classified as small (<3.9 cm), medium (4.0–5.9 cm) or large (>6 cm). Five sea urchin species occur at the study site, which include Hemicentrotus pulcherrimus, Toxopnuestes pileolus, D. savignyi, D. setosum, and H. crassispina. Of these, H. pulcherrimus was cryptic, usually found under boulders and was observed along the transect only once, whereas T. pileolus were scattered widely, and only thirteen individuals were found during the monitoring period. Given the scarcity of T. pileolus and H. pulcherrimus, they were not included in the formal analysis. Sea urchin biomass was estimated from sea urchin size-weight relationship determined by fitting a generalized linear model (GLM) on the weight and size from 10 randomly collected individuals of D. savignyi, D. setosum and H. crassispina from the shallow and deep transects.

Quadrat photos were downloaded from the camera and 50 points were randomly plotted on each photo across a 10 × 10 grid via R (R Core Team, 2022) using the magick (Ooms, 2021) and imager (Barthelme et al., 2021) packages. Each point was identified and classified whether they were macroalgae, coralline algae (i.e., encrusting, geniculate), turf algae, substrate (i.e., rock, sand, silt), or other (i.e., debris, benthic animals). Benthic elements not associated with benthic cover (i.e., other) were excluded from the analysis.

Environmental conditions were recorded using data loggers. For each habitat, one wave height logger (Infinity-WH AWH-USB; Alec Electronics Co., Kobe, Japan) was deployed at a point midway between the deep and shallow transects. For each deep and shallow belt transect, one depth logger (HOBO U20; Onset Computer Corp., Bourne, MA, USA) and one temperature/light logger (HOBO Pendant MX2202; Onset Computer Corp., Bourne, MA, USA) was installed. The instruments recorded environmental data at 10-min intervals for 20–25 days each month before retrieval and data offloading. The monthly daily averages for wave height (m) and temperature (°C) were calculated for each month. The monthly average of the integrated photosynthetic photon flux density (PPFD, mol m−2 day−1) was also calculated for each month.

Sea urchin mark-recapture experiment

To measure sea urchin movement patterns within a 24 h period, mark-recapture experiments were conducted in late August to early September 2020 on the two most abundant sea urchin species at the study sites; the long-spined black sea urchin, D. setosum and the purple sea urchin, H. crassispina. Whereas D. setosum occurred in groups of 2–30 individuals, H. crassispina was solitary. Twenty individuals each of D. setosum and H. crassispina from each habitat were selected, while ensuring that the individuals were separated by at least 5 m. The initial location of all individuals was marked by placing numbered buoys. A numbered plastic T-bar tag was inserted into the inter-ambulacral region using a tagging gun with a 0.5 mm diameter needle (Rodríguez-Barreras & Sabat, 2015), and each tagged individual was treated as one trial. All tagged sea urchins were returned to their original position and the number of individuals which composed the group at the initial location, were also recorded. The initial capture and tagging activity was conducted between 06:00 to 07:00, labelled as “start” and assigned a linear distance of 0 m. The first recapture occurred after 12 h (i.e., 18:00) and the second recapture was after another 12 h (i.e., 6:00–7:00 on the next day). About 5–10 min was allocated for recapturing the tagged sea urchins via a circular search pattern by snorkeling. Once a tagged individual was found, the marked buoy was moved to the new location and the linear displacement was measured. Those not found during the recapture were labelled as lost and excluded from the analysis.

Benthic rugosity measurements

Rugosity can be used as a proxy to describe substrate complexity, where a flat surface would have a value of one and higher values can describe a more complex benthic structure. Benthic rugosity was measured by laying a 5 m length of stainless-steel chain with 1 cm links along the contour of the rocky reef. A transect tape was laid beside the chain to measure the straight-line length the chain end-points (Risk, 1972; Trebilco et al., 2015). The straight-line chain length (5 m) was divided by the overlaid length of the chain to estimate rugosity. A total of 35 and 25 measurements were done at the deep and shallow areas, respectively in the IH and 24 and 36 were done at the deep and shallow areas, respectively of the VH.

Data analysis

Environmental data and phenology

Poor weather conditions and instrument error with the temperature/light loggers led to several months with missing data (Fig. S2). For example, in the VH, from July–December 2021 (i.e., missing data in the shallow VH from July to December and missing data in the deep VH from July, November, and December), and in January and April 2021, in both sites. The missing temperature and light data were imputed by fitting a generalized additive model (GAM) (Eq. (1)) to the datasets. The monthly daily averaged wave heights were determined assuming a gamma distribution with a log link-function, whereas the monthly daily averaged light and temperature data sets were determined assuming a gaussian distribution with a log and identity link-function, respectively. Due to excess zeroes in observations, the sea urchin density dataset was analyzed assuming a hurdle negative binomial distribution with a log link-function, whereas the sea urchin biomass dataset was analyzed assuming a hurdle gamma distribution with a log link-function. For the percent benthic cover, benthic elements were modelled separately due to different patterns in spatial distribution. Due to excess zeroes, coralline, macroalgae and turf elements were analyzed assuming a zero-inflated beta distribution with a logit link-function, whereas the substrate element was analyzed assuming a zero-one inflated beta distribution with a logit link-function. The smoothing function for all GAMs was a cubic regression spline.

(1) μ=g(y,θ)y=f(s(x)+β1+β2)

β∼Normal(0,0.5)θ∼Student3(0,1)

Here, g is the distribution, f is the link-function, s is a smoothing function for time x, μ is the location and θ is the scale of the g, and β is the coefficient for the covariates (i.e., habitat and transect). For the average daily wave height dataset, the predictor variables were the time and location (two factor levels: IH, VH). For the average daily light and temperature, percent benthic cover, sea urchin density, and biomass datasets the predictor variables were the time, location (two factor levels: IH, VH), and transect (two factor levels: deep, shallow). For all model priors, the β coefficients were assigned a normal prior with a location of 0 and a scale of 0.5 while θ were assigned Student’s t prior with three degrees-of-freedom, a location of 0, and scale of 1.

Benthic rugosity

The difference in benthic rugosity between IH and VH was analyzed using a generalized linear model (GLM) assuming a Gamma distribution ( Γ()) with a log link-function (Eq. (2)). The response variable was the benthic rugosity and the predictor variables were the habitat (two factor levels: IH, VH) and transect (two factor levels: deep, shallow).

(2) y=Γ(μ,θ)μ=exp(α+βx)

β∼Normal(0,0.5)θ∼Student3(0,1)

Here, the y is the measured benthic rugosity across the deep and shallow transects of the IH and VH; α and β are the model coefficients; x is the predictor variable. The parameters μ and θ are the location and scale of the Gamma distribution. For the model prior, the β coefficients were assigned a normal prior with a location of 0 and a scale of 0.5 while θ was assigned a Student’s t prior with three degrees-of-freedom, a location of 0, and scale of 1.

Sea urchin size-class and microhabitat preference

The size-class and preferred microhabitat of D. savignyi, D. setosum and H. crassispina was analyzed using a GAM assuming a beta distribution with a logit link-function to elucidate the relationship between the proportion of sea urchin species (y) occurring in a particular microhabitat in the IH and VH, and their corresponding size-class during the monitoring period. The model is similar to Eq. (1) and the smoothing function for the GAM was a cubic regression spline.

The predictor variables were habitat (two factor levels: IH, VH), transect (two factor levels: deep, shallow), species (three factor levels: D. savigny, D. setosum, H. crassispina), microhabitat (three factor levels: pit, crevice, free-living), size-class (three factor levels: small, medium, large). The β coefficients were assigned a normal prior with a location of 0 and a scale of 0.5 while θ were assigned a Student’s t prior with three degrees-of-freedom, a location of 0, and scale of 1.

Sea urchin movement patterns

The linear displacement by D. setosum and H. crassispina over a 24 h period during the mark-recapture experiment was analyzed using a GLM, assuming a hurdle-gamma distribution (Lewin et al., 2010) with a log link-function (Eq. (3)).

(3) y=(1−π)Γ(0,θ)+πΓ(μ,θ)μ=xβlogπ1−π=xα

β∼Normal(0,1)θ∼Student3(0,1)θθ,trial∼Student3(0,2)

Here, y is the sea urchin displacement over the mark-recapture period; π is the probability of a non-zero value; α and β are the model coefficients; x is the predictor variable. The parameters μ and θ are the location and scale of the Gamma distribution, respectively. The predictor variables were urchin species (two factor levels: D. setosum and H. crassispina), habitat (two factor levels: IH, VH), recapture period (two factor levels: first recapture, and second recapture), and the experimental trial was treated as a random intercept. The β coefficients were assigned a normal prior with a location of 0 and a scale of 0.5 while θ was assigned a Student’s t prior with three degrees-of-freedom, a location of 0, and scale of 2.

The change in sea urchin group size was analyzed using a Bayesian GLM which assumed a negative binomial distribution with a log-link function (Eq. (4)). For this analysis, H. crassispina was excluded since it was solitary throughout the mark-recapture period.

(4) y=NegBin(μ,θ)μ=exp(β1+β2)

β1~Student3(0,θβ1)β2~Student3(0,θβ2)θ~Student3(0,θθ)θβ1,trial~Student3(0,2)θβ2,trial~Student3(0,2)θθ,trial~Student3(0,2)

where, y is the number of D. setosum individuals together with the tagged sea urchin during the start, first recapture, or second recapture period; The parameters μ and θ are the location and scale of the negative binomial distribution. The θ and β coefficients were assigned Student’s t priors with three degrees-of-freedom, a location of 0 and a scale of 2. The predictor variables were the recapture period (three factor levels: start, first recapture, second recapture), and habitat (two factor levels: IH, VH), while the period and habitat were nested in the varying intercept, experimental trial.

All analyses were conducted in R version 4.1.2 (R Core Team, 2022). Bayesian methods were used for all models through the brms package (Bürkner, 2017, 2018) and assigned weakly informative priors (Gelman, Simpson & Betancourt, 2017), where each model was ran with four Markov chains with 4,000 iterations per chain. The chains and posterior distributions were assessed visually for convergence (see model validations in Figs. S3–S17). All models were compared with a similar model having the same distributional parameters but with no explanatory variables (null model) using the difference in the expected log pointwise predictive density (ELPD) of the leave-one-out cross validation (LOO) (Table 1) (Vehtari, Gelman & Gabry, 2017). As the data were not normally distributed, we reported the conditional means that were estimated by the Bayesian models.

Table 1 Model comparison between all models and their equivalent null model.

Model	ELPD difference	Standard error	
(A) Environmental condition models			
Wave height GAM	31.98	4.88	
Light (PPFD) GAM	17.96	6.66	
Temperature GAM	105.15	6.02	
(B) Benthic terrain model			
Rugosity GLM	22.86	5.30	
(C) Benthic cover models			
Coralline GAM	52.82	5.86	
Macroalgal GAM	41.28	4.44	
Substrate GAM	57.95	5.46	
Turf GAM	18.57	4.91	
(D) Sea urchin density models			
D. savignyi density GAM	7.31	4.99	
D. setosum density GAM	20.06	7.64	
H. crassispina density GAM	46.83	6.24	
(E) Sea urchin biomass models			
D. savignyi biomass GAM	1.96	4.10	
D. setosum biomass GAM	4.14	4.75	
H. crassispina biomass GAM	14.04	6.08	
(F) Microhabitat preference model			
Microhabitat GAM	88.47	13.32	
(G) Sea urchin tagging models			
Linear displacement GLM	73.57	8.20	
Group-size GLM	19.37	5.98	
Note:

The values are the absolute difference in the expected log point-wise predictive density (ELPD) of the leave-one-out cross-validation (LOO) between all models and their equivalent null model. The results indicate support for the full model over the null.

Results

The monthly daily averaged wave heights in the IH and VH over the study period ranged from 0.01 to 0.02 m and 0.03 to 0.09 m, respectively (Table S1). The highest waves were recorded from autumn until spring, and it was during this time (July–December 2021) when the temperature/ light loggers were damaged in the shallow VH resulting in missing data. The lowest wave heights were during the late-spring and summer months (Fig. 2A). The light levels in the IH and VH had monthly daily average photosynthetic photon flux densities (PPFD) ranges of 1.47 to 16.27 mol m−2 day−1 and 1.82 to 12.86 mol m−2 day−1, respectively (Fig. 2B and Table S2). Meanwhile, the monthly daily average temperature at IH and VH during the study period ranged from 13.24 to 27.05 °C and 13.51 to 27.19 °C, respectively (Fig. 2C and Table S3). The difference in the expected log point-wise predictive density (ELPD) of the leave-one-out cross validation (Table 1) indicates support for the generalized additive models (GAM) applied to the environmental data over their equivalent null models (Vehtari, Gelman & Gabry, 2017).

Figure 2 The environmental data recorded by instruments during the study.

The monthly daily average (A) wave heights, (B) light levels (PPFD), and (C) temperature from the isoyake habitat and vegetated habitat. The points are the observations, the solid and dashed lines are the expectations of the generalized additive models and the shaded regions indicate the 95% highest density credible interval.

The substrate at the VH is composed of rocks and large boulders while the IH benthic composition was mainly small flat rocks and silt. The rugosity GLM shows lower mean rugosities for the deep (1.16) and shallow (1.23) areas for IH, compared to the deep (1.38) and shallow (1.37) areas of VH (Fig. 3 and Table S4). Both habitats experienced macroalgal blooms from late winter until early autumn (Fig. 4A and Table S5). In general, peak macroalgal blooms occurred in February for both the deep (11.62%) and shallow (31.10%) IH while peak macroalgal blooms occurred in April (13.44%) for the deep VH and in May (33.36%) for the shallow VH (Fig. 5). The macroalgae composing the blooms in the IH are mostly brown seaweeds (i.e., Colpomenia sinuosa, Hydroclathrus clathratus and Sargassum horneri) whereas algae in the VH is composed of green (i.e., Codium spp.), brown (i.e., C. sinuosa, H. clathratus, Dictyota spp., Dictyopteris spp., Padina spp., Sargassum spp.) and red algae (i.e., Asparagopsis taxiformis, Martensia jejuensis). Although the persistence of macroalgal cover (>1%) at the shallow VH (3.52%) and deep IH (1.10%) occurred until September while the macroalgae at deep VH (2.78%) and shallow IH (1.82%) lasted until August. The relative importance of habitat type and transect depth shows that the macroalgal state was similar in both habitats (Table S6). Outside the macroalgal bloom season, the IH was mostly bare substrate (i.e., rocks and silt), and the substrate cover state persisted throughout the study period (IH deep: 0.446, IH shallow: 0.258, Table S6). The substrate cover state also persisted in the deep VH (0.242), while the shallow VH had a lower overall rate of exposed substrate (0.054).

Figure 3 The benthic rugosity at the isoyake habitat and vegetated habitat.

The pale-colored points indicate the observations while the solid points are the mean expected rugosity of the generalized linear model and the vertical lines are the 95% highest density credible interval.

Figure 4 The results of the benthic quadrat monitoring activity.

The benthic monitoring study showing the (A) percent benthic cover, (B) sea urchin density, and (C) sea urchin biomass from the deep and shallow transects of the isoyake and vegetated habitats. The points are the monthly observations, the solid, dashed, and dotted lines are the expectations of the generalized additive models and the shaded regions indicate the 95% highest density credible interval.

Figure 5 Quadrat photos from the isoyake habitat (IH) and vegetated habitats (VH) in seasons with (A) high and (B) low algal cover.

Quadrat photos showing the peak algal cover for (A1) deep IH and (A2) shallow IH in February and in the months of April and May for (A3) deep VH and (A4) shallow VH, respectively. The same quadrats during the low macroalgal algal cover period (<1%) occurring in August for (B2) shallow IH and (B3) deep VH and in September for (B1) deep IH and (B4) shallow VH.

Among the three sea urchin species, D. setosum had similar patterns of densities across all depths in both study sites (Table S6), while H. crassispina had an affinity for the shallow VH and IH, followed by the deep VH and deep IH. For D. savignyi, its distribution pattern was less clear but its preference for the shallow VH was higher compared to the deep IH. Population-wise, sea urchin densities (Fig. 4B and Table S7) indicated that D. setosum was the most common in the deep (1.48 indiv. m−2) and shallow areas (1.32 indiv. m−2) of IH and in the deep VH (2.95 indiv. m−2). In the shallow VH, H. crassispina was most common (3.02 indiv. m−2). The least abundant species in all sites was D. savignyi (<1 indiv. m−2). The sea urchin biomass (Fig. 4C and Table S8) in the IH was generally low. In the deep IH, H. crassispina had the lowest biomass among the three species (1.27–1.63 g m−2), followed by D. savignyi (2.68–3.85 g m−2) and D. setosum (3.42–4.52 g m−2), while In the shallow IH, the three species had similar biomass. In the shallow VH, D. savignyi had the highest biomass (4.84–7.17 g m−2), followed by D. setosum (3.40–4.50 g m−2) and H. crassispina with the least (1.50–1.79 g m−2).

Sea urchin size-class and microhabitat preference

The sea urchins were distributed widely across the two study sites and had distinct microhabitat preferences during the study period (Fig. 6 and Table S9). At the IH, greater than 50% of medium-sized D. setosum occurred as free-living during the autumn and early winter months of 2020 and during the summer-autumn months of 2021 (Figs. 6A and 6B). However, the preference for crevices among D. setosum increased from the winter months of 2020 until mid-spring of 2021. The same patterns were observed for the deep and shallow transects of IH. For D. savignyi, few sea urchins were recorded at IH. In the deep area, an increasing trend was found for small individuals preferring crevices from spring until summer 2021, while less than 50% were free-living. In the shallow IH, the occurrence of D. savignyi in crevices and as free-living was below 25%. For H. crassispina, sea urchins preferred mostly crevices in the deep IH. Free-living H. crassispina in the deep IH occurred only in September of 2020 and 2021 (13.94% and 10.86%, respectively). The H. crassispina found in the shallow IH was found in all microhabitat types and almost exclusively composed of medium-sized individuals. However, free-living individuals were not recorded from August to December 2021.

Figure 6 Sea urchin size-classes and their microhabitat preference in the (A) deep and (B) shallow isoyake habitats and the (C) deep and (D) shallow vegetated habitats.

The columns show the sea urchin species and the rows show the microhabitat type. The points are the observations, the solid, dashed and dotted lines are the expected sea urchin occurrence rate of the generalized additive model and the shaded regions indicate the 95% highest density credible interval.

In the deep VH, D. setosum was most abundant (Fig. 6C), composed mainly of small and large-sized individuals and followed similar seasonal patterns of microhabitat preference as those in IH. In the shallow VH, all size classes of D. setosum mostly preferred crevices while few medium and large-sized individuals were free-living from spring to late summer of 2021. For D. savignyi, small and large individuals preferred crevices while small individuals were free-living in the deep VH. In the shallow VH, medium and large individuals preferred crevices while large individuals were free-living. For H. crassispina, small sea urchins mostly preferred crevices together with a few large individuals in the deep VH throughout the study period. Although free-living individuals were found in the spring of 2021, H. crassispina in the shallow VH generally preferred either pits or crevices throughout the study period.

Sea urchin movement patterns

There was a difference between the movement patterns of H. crassispina and D. setosum across the IH and VH (Fig. 7A), with a clear pattern that shows increasing linear distance after the second recapture compared to the first recapture. On average, both species displaced further in the IH than at the VH in the second recapture (Table S10).

Figure 7 The results of the sea urchin mark-recapture experiment.

(A) Shows the average linear displacement by the tagged Diadema setosum and Heliocidaris crassispina across the first and second recapture activity, while (B) is the change in the D. setosum group composition together with the tagged individuals. All tagged H. crassispina were solitary throughout mark-recapture experiment and were excluded from the group composition analysis. The pale-colored points indicate the observations while the solid points are the mean expected values of the generalized linear model. The vertical lines indicate the 95% highest density credible interval.

Both species also had distinct social behaviors because H. crassispina was always solitary in its crevice or burrow and was not found together with other conspecifics when it was outside. In contrast, D. setosum was always observed in groups. The mark-recapture experiment shows that D. setosum group composition varied widely in the VH compared to the IH (Fig. 7B). The group-size GLM (Table S11) shows that there was a decrease in the average number of individuals in the first recapture in both IH (from 4.38 to 2.16 indiv.) and VH (from 7.73 to 5.55 indiv.) while an increase was observed for the group size in IH (from 2.16 to 5.29 indiv.) but not for VH (from 5.55 to 4.80 indiv.), after the second recapture.

The relatively large displacement and the changing group size by D. setosum caused difficulty in locating the tagged individuals during the recapture periods leading to losses (Table 2). Following the second recapture in VH, 25% (n = 5 indiv.) of tagged D. setosum were lost. At IH, 30% (n = 6 indiv.) were lost during the first recapture and 50% (n = 10 indiv.) were subsequently lost after the second recapture period. For H. crassispina, eighteen individuals in VH remained in their initial positions throughout the experiment while two were lost after the second recapture period. All H. crassispina were located in the IH.

Table 2 The sea urchin recapture rate and loss rate from the mark-recapture experiment conducted at the isoyake habitat (IH) and vegetated habitat (VH).

Habitat	Species	Survey	N	Recapture rate (%)	Loss rate (%)	
Isoyake	D. setosum	Start	20	100	0	
Isoyake	D. setosum	1st recapture	14	70	30	
Isoyake	D. setosum	2nd recapture	10	50	50	
Isoyake	H. crassispina	Start	20	100	0	
Isoyake	H. crassispina	1st recapture	20	100	0	
Isoyake	H. crassispina	2nd recapture	20	100	0	
Vegetated	D. setosum	Start	20	100	0	
Vegetated	D. setosum	1st recapture	20	100	0	
Vegetated	D. setosum	2nd recapture	15	75	25	
Vegetated	H. crassispina	Start	20	100	0	
Vegetated	H. crassispina	1st recapture	20	100	0	
Vegetated	H. crassispina	2nd recapture	18	90	10	
Note:

The mark-recapture experiment involved tagging Diadema setosum (IH: 20 indiv., VH: 20 indiv.) and Heliocidaris crassispina (IH: 20 indiv., VH: 20 indiv.). The species H. crassispina had higher recapture rates because it was less mobile while D. setosum had a higher loss rate, due to its higher mobility.

Discussion

For over a year we compared adjacent habitats with contrasting ecosystem states and evaluated how their environmental conditions and benthic phenology varied and how different sympatric sea urchins behaved across both sites. Both sites differed in hydrodynamic conditions as the VH is more exposed while the IH is generally sheltered (Fig. 2A), despite having similar wind fetch patterns (Fig. S1). This has large implications for water visibility and sedimentation rates because the deeper parts of IH is generally turbid leading to greater light attenuation (Fig. 2B), possibly affecting benthic communities (Fig. 4A). Although we did not measure productivity, it is known that a stable water column facilitate phytoplankton blooms (Matsumoto et al., 2021), especially in bays and sheltered areas. Alternatively, vertical mixing due to tidal fluctuations can also resuspend sediments and lead to turbidity (Wang, 2002).

The macroalgal bloom occurring in spring until summer across both sites is a period of high primary productivity and algal food availability for benthic communities. Interestingly, the deep and shallow IH reached peak macroalgal cover three months earlier compared to the shallow VH but also ended earlier while macroalgal cover persisted until early autumn in the shallow VH. For both habitats, it appears that the benthic macroalgal bloom may be controlled by seasonal environmental cues attributed to changes in environmental variables such as temperature and light (Yoshida, Yoshikawa & Terawaki, 2001; Toste et al., 2003; Martínez, Pato & Rico, 2012; Gauna, Cáceres & Parodi, 2013). For example, daylength caused an increase in the growth of the frond length, biomass, branching rate for Dictyota dichotoma (Gauna, Cáceres & Parodi, 2013), a species common in the VH but not in the IH. However, we suspect that the asynchronous peak benthic cover and composition of the macroalgal blooms may be due to factors such as nutrient supply. Specifically, the IH is in a semi-enclosed area prone to the effects of terrestrial run-off. Benthic communities close to the source of land-based pollution are known to have lesser diversity and composed of species with simpler thalli forms with relatively short life histories (Littler & Murray, 1975) that benefit from rapid uptake of the excess dissolved inorganic nutrients. This may help explain why the macroalgal bloom in the IH is predominantly composed of Colpomenia sinuosa and Hydroclathrus clathratus. Furthermore, the substrate type may also influence the persistence of benthic algae because hard substrates are better at supporting a high diversity of macroalgae and associated fauna while soft substrates support benthic microalgal communities and burrowing invertebrates (Chenelot, Jewett & Hoberg, 2011) and do not provide a stable attachment for seaweed holdfasts. In general, the shallow VH has a wider variety of primary producers composed of seasonal macroalgae and perennial coralline and turf algae, while the entire IH and the deep VH only has a seasonal macroalgal bloom and some coralline algae, while the substrate is generally bare.

Sea urchin size-class and microhabitat preference

Sea urchin microhabitat preference may have been associated with changes in environmental conditions because the gradual decline in the occurrence of free-living D. setosum from autumn until spring and the increase in preference for crevices during the same period (Figs. 6A–6C) closely coincides with high wave action around those months (Fig. 2A). Shelter-seeking behaviors may be a response to strong hydrodynamic forces because during the calmer months in summer, the occurrence of free-living sea urchins gradually increased. It is known that high water motion reduces urchin movement and foraging behavior due to the threat of dislodgement (Kawamata, 1998; Siddon & Witman, 2003; Cohen-Rengifo et al., 2018). Urchins respond to increasing water motion by escaping from exposed areas and changing their outline to a more streamlined shape (Cohen-Rengifo et al., 2018) or remain sheltered until conditions improve (Yusa & Yamamoto, 1994; Tamaki, Muraoka & Inoue, 2018). In addition, D. savignyi, a closely related Diadematid species shows a similar pattern in preferring crevices in the deep IH but only few individuals were found. In the entire VH, D. savignyi were mostly crevice dwellers throughout the study period. We speculate that although Diadematid urchins (i.e., D. setosum and D. savignyi) occupy the same guild, D. setosum was a better competitor for resources resulting in a skewed population in favor of D. setosum. For H. crassispina, this species generally prefers crevices throughout both habitats except in calm conditions where they may leave their crevices as in the shallow IH but also to a lesser extent in the shallow VH. Very few sea urchins of all species were found in pits along the deep VH and IH (Figs. 6A and 6C) because pit microhabitats were scarce in these areas. Overall, it was clear that the IH was mostly composed of small and medium sized urchins while there was a higher chance of observing larger individuals in the VH, especially for D. savignyi and D. setosum, probably due to the food availability. A study on interactions between Echinometra mathaei, D. savignyi and D. setosum in a coral reef shows that among Diadematid urchins, the body size characteristics and sedentary habits of D. savignyi enabled it to outcompete D. setosum for microhabitat refuge while E. mathaei was the top competitor for crevice microhabitats due to settlement success while benefiting greatly from living in crevices because of high predation pressure compared to Diadematid urchins (McClanahan, 1988). In our study, we suspected that the sedentary habit of D. savignyi was a disadvantage particularly in areas where resources were scarce. In these conditions, active foraging is better suited to maximize encountering food and refuge. Since our surveys were done in the daytime, we were not able to observe urchin foraging at night when they are most active (Tuya, Martin & Luque, 2004). Among the three species, H. crassispina was the exception, because it primarily preferred crevices while rates of being free-living occurred only in the shallow IH where wave action was less. A good example of this scenario is in Hong Kong where H. crassispina form destructive feeding fronts in sheltered bays (Urriago Suarez et al., 2021).

According to historical records, D. setosum was implicated in the loss of seaweed beds in Japan since the late 1800’s (Fujita, 2010) and are regularly culled as a first step in the rehabilitation of barren areas (Nanri et al., 2011; Ohmura, Watanabe & Fujita, 2011). Unlike other species, Diadematid urchins have lesser commercial value in Japan due to the higher ratios of bitter tasting compounds in their gonads (Kaneko et al., 2009) while their optimal season occurs only in June (Kaneko et al., 2012). In our study, sea urchin densities in both habitats were low relative to thresholds necessary to trigger a forward phase shift (Fig. 8), but were enough in some months to maintain the barren state (Fig. 8, shallow VH, August–October) or even support recovery (Fig. 8, deep IH) (Ishikawa, Maegawa & Kurashima, 2016; Kriegisch et al., 2016; Ishikawa & Kurashima, 2020). Sea urchin barrens are known for their inherent stability since thresholds for macroalgal recovery (reverse shift) is different from the one which initiated the shift (forward shift) (Filbee-Dexter & Scheibling, 2014). The environmental conditions in the IH may be contributing to unfavorable conditions for algal growth despite having lower urchin densities compared to the VH.

Figure 8 Sea urchin densities (indiv. m−2) that describe a discontinuous phase shift.

Sea urchin densities have important thresholds that indicate the likelihood of a phase shift (see, Ishikawa, Maegawa & Kurashima, 2016; Kriegisch et al., 2016; Ishikawa & Kurashima, 2020). The red dotted line indicates sea urchin densities that cause a phase shift (about 8 indiv. m−2). The orange dotted line indicates sea urchin densities that maintain the barren state (about 4 indiv. m−2). The green dotted line indicates sea urchin densities that allow the recovery of seaweed beds (2 indiv. m−2 or less).

A meta-analysis of studies on macroalgal bed-to-barren regime shifts found that sea urchin biomass over 668 g m−2 led to overgrazing while about 34–71 g m−2 allows for macroalgal recovery (Ling et al., 2015). Based on these estimates, the sea urchin biomass in our transects were at the lower limits of those that were reported. However, the sea urchin biomass in our study could be underestimating the actual values because sea urchin weights were inferred by sampling a few individuals in the spring. Sea urchin reproductive cycles are marked by distinct events that affect their overall biomass. For example, gonadal development and maturation and the high availability of algal food lead to increasing body mass in the winter and spring, while spawning and the low macroalgal diversity in the summer to autumn lead to lower body weights (Kaehler & Kennish, 1996; Horii, 1997; Bronstein, Kroh & Loya, 2016; Urriago et al., 2016). Hence, we suggest considering sea urchin density as an indicator for estimating the risk of phase shifts as biomass is dependent on season. The Arikawa Bay Fisheries Cooperative that manages the fisheries rights of the study area has indicated that sea urchin culling and harvesting has not occurred at our study sites prior to or during this study.

Sea urchin movement patterns

In terms of sea urchin movement patterns, we demonstrated how sympatric sea urchin species behaved differently in adjacent habitats. Generally, both taxa were more active and displaced farther over a 24 h period in the IH where wave action and benthic rugosity was low, compared to the VH. The displacement GLM showed a higher displacement occurring after the second recapture compared to the first recapture period (Fig. 7A), indicating both species were nocturnal. A study on Diadema antillarum, show nocturnal foraging behaviors and site fidelity among the tagged urchins (Tuya, Martin & Luque, 2004). However, we were not able to determine whether D. setosum displayed homing behaviors during our tagging study. We interpret the sea urchin movements as a response to wave action and as foraging behavior. The lower wave action and the relative food scarcity prompted urchins to forage more in the IH, whereas the necessity to forage further was less at VH where waves were stronger and food was relatively abundant. A number of studies document the flexibility of sea urchin diets to include detritus and animal matter when algal food is low (Freeman, 2003; Wangensteen et al., 2011; Rodríguez-Barreras et al., 2015; Umezu et al., 2017; Camps-Castellà, Romero & Prado, 2020). When food was abundant, sea urchins have been known to subsist on drift algae (Kelly, Krumhansl & Scheibling, 2012; Kriegisch et al., 2019; Rennick et al., 2022), reducing the need to forage far from their refuge which indirectly reduces feeding pressure on the existing seaweed beds. It is also known that the threat of predation induces alarm responses among several species of sea urchins (Snyder & Snyder, 1970; Campbell et al., 2001; Morishita & Barreto, 2011). Recent studies have found that sea urchin response to predator cues may vary depending on species. For example, when a dead conspecific was present, Strongylocentrotus intermedius formed aggregations while Mesocentrotus nudus responded by leaving the vicinity (Zhadan & Vaschenko, 2019). There is also some evidence that for species like Echinometra mathaei (Hart & Chia, 1990) and H. crassispina (Belleza et al., 2021), starvation weakens the anti-predator response and prioritizes foraging behaviors, but not in M. nudus (Chi et al., 2021).

As for the patterns observed with the changes in urchin aggregations, the group-size GLM shows that all H. crassispina remained solitary throughout the study. There was a net decrease in D. setosum group-size in both habitats after the first recapture and a subsequent increase in group-size after the second recapture in the IH but not in the VH (Fig. 7B). Furthermore, it is important to note that D. setosum group sizes in the IH were similar while there were large variations in group composition among experimental trials in the VH. We suspect that the high availability of food and refuge in the VH supported higher recruitment survivability and safety from predation, although the level of predation pressure on sea urchins in both habitats remains unclear. It is known that aggregation behavior is one strategy to deter predators (Garnick, 1978) and are maintained by close spine contact between members of the group while increased spine movement indicates defensive activity against a potential predator (Morishita & Barreto, 2011). Nevertheless, tagging activities tend to disrupt aggregations when an individual is taken from the group. The tagging procedure was invasive and involved puncturing a 0.5 mm hole on the sea urchin’s test at the inter-ambulacral region to insert the tag. Even though the tagging injury is small, urchin coelomic fluids may still leak out and evoke a flee response from the surrounding urchins when the tagged urchin is returned to the group (Campbell et al., 2001; Morishita & Barreto, 2011; Spyksma, Shears & Taylor, 2020). This may help explain the reduction in group size in the first recapture. After the second recapture, the tagged D. setosum may have partially healed and retained some group members or have encountered other aggregations while foraging at night. This phenomenon has been observed in the Solomon Islands where D. setosum and D. savignyi aggregations are constantly changing in size, and aggregations of both species often combine (Pearse & Arch, 1969). In the IH, we expect sea urchin aggregations to encounter other groups easily in the flat terrain but not in the VH where abundant crevices and wave action results in more dynamic group interactions. Furthermore, the relative abundance of food in the VH reduces the necessity to forage further.

Past records and future implications

While the importance of sea urchins as key drivers of ecosystem states is generally recognized, it is equally important to note environmental changes that occurred in recent years. A fisheries survey in 1991 shows that VH was once a vibrant algal bed that included five species of Sargassum, three subtropical kelp species and about four species of abalone and several commercially important fish species (Nagasaki Prefecture Fisheries Development Association, 1991). There were no official records of surveys done at IH but remote sensing maps available through the Geospatial Information Authority of Japan (https://www.gsi.go.jp/johofukyu/johofukyu210322.html), shows that IH was once possibly vegetated in the late 1960’s prior to coastal armoring and the construction of the concrete jetty in the early 1970’s (Fig. 9), given its proximity to areas previously surveyed (Nagasaki Prefecture Fisheries Development Association, 1991). Urbanization and coastal development are known anthropogenic stressors that affect shallow subtidal ecosystems (Kevekordes, 2001; Coleman et al., 2008; Coleman & Wernberg, 2017). In particular, coastal armoring in Japan has been used extensively as a response to coastal erosion, sea level rise, and storm surge (Koike, 1996). Ironically, a trade-off occurs as natural habitats are lost for coastal protection. In this case, we speculate that coastal development has contributed to the loss of a seaweed habitat by altering the area’s hydrodynamic conditions leading to increased sedimentation (Airoldi, 1998, 2003). Furthermore, the average winter temperature in Arikawa Bay appear to have increased and remained elevated since 2018 (Fig. 10, GN Nishihara, unpublished data; Supplemental files: Raw table 11). Higher winter temperatures mean sea urchins (Ishikawa & Kurashima, 2020) and herbivorous fish (Yamaguchi et al., 2006; Noda et al., 2016) are able to remain active and grazing pressure may persist throughout the year.

Figure 9 Comparison of the study site pre- and post-coastal development.

Maps showing the coastline of the study site, (A) before coastal armoring in the early 1970’s and (B) present-day. The red dots are probable seaweed habitats that were potentially affected during the construction of the concrete jetty. Image credit: the Geospatial Information Authority of Japan (https://www.gsi.go.jp/johofukyu/johofukyu210322.html).

Figure 10 The annual average winter temperatures (i.e., December, January, February) in the vegetated habitat of Arikawa Bay since 2018.

The points are the mean values while the vertical bars are the standard deviation.

Conclusions

Our study highlights the similarities and differences in the behavior and distribution of sympatric sea urchins, Diadema savignyi, D. setosum and Heliocidaris crassispina in adjacent habitats under natural conditions. We found that, (1) sea urchins show an increased preference for sheltered habitats such as crevices and burrows during the months with high wave action while the incidence of free-living sea urchins increased during the calmer months. (2) In areas where benthic rugosity and wave action are low, sea urchins tend to displace further.

Supplemental Information

Supplemental Information 1 Supplementary figures.

Document containing (A) the mean wind fetch for the isoyake, (B) the average daily wave heights, PPFD and temperature conditions of the vegetated and isoyake habitats, and (C) the figures for all model validations, each showing their kernel density estimates of the observations and predictions and rank plots for the MCMC draws.

Click here for additional data file.

Supplemental Information 2 The result of the Bayesian GAM applied to the monthly daily average wave height dataset.

The mean and 95% highest density credible interval for the expectations of the generalized additive model (GAM) of the monthly daily average wave heights (m) in the isoyake and vegetated habitat.

Click here for additional data file.

Supplemental Information 3 The result of the Bayesian GAM applied to the monthly daily average PPFD dataset.

The mean and 95% highest density credible interval for the expectations of the generalized additive model (GAM) applied to the monthly daily average photosynthetic photon flux density (PPFD, mol m−2 day−1) in the deep and shallow transects of the isoyake and vegetated habitat.

Click here for additional data file.

Supplemental Information 4 The result of the Bayesian GAM applied to the monthly daily average temperature dataset.

The mean and 95% highest density credible interval for the expectations of the model on the monthly daily average temperature (°C) in the deep and shallow transects of the isoyake and vegetated habitat.

Click here for additional data file.

Supplemental Information 5 The result of the Bayesian GLM applied to the benthic rugosity dataset.

The mean and 95% highest density credible interval for the expectations of the model on benthic rugosity in the deep and shallow transects of the isoyake and vegetated habitat.

Click here for additional data file.

Supplemental Information 6 The result of the Bayesian GAM applied to the algal benthic cover monitoring dataset.

The mean and 95% highest density credible interval for the expectations of the model on the percent benthic cover in the deep and shallow transects of the isoyake and vegetated habitat.

Click here for additional data file.

Supplemental Information 7 The relative importance for the parameters of all models and their distributions.

The relative importance is the ratio of the absolute values of the coefficients for the model’s main effects excluding the covariate interactions and effects of the time smoother, in the case of Generalized Additive Models (GAM).

Click here for additional data file.

Supplemental Information 8 The result of the Bayesian GAM applied to the sea urchin density dataset.

The mean and 95% highest density credible interval for the expectations of the model on the sea urchin density (indiv. m−2) in the deep and shallow transects of the isoyake and vegetated habitat.

Click here for additional data file.

Supplemental Information 9 The result of the Bayesian GAM applied to the sea urchin biomass dataset.

The mean and 95% highest density credible interval for the expectations of the model on the sea urchin biomass (grams m−2) in the deep and shallow transects of the isoyake and vegetated habitat.

Click here for additional data file.

Supplemental Information 10 The result of the Bayesian GAM applied to the sea urchin size-class and microhabitat preference dataset.

The mean expected occurrence rate and 95% highest density credible interval for the expectations of the model on the sea urchin size-class and microhabitat preference in the deep and shallow transects of the isoyake and vegetated habitat.

Click here for additional data file.

Supplemental Information 11 The result of the Bayesian GLM applied to the sea urchin linear displacement dataset.

The mean and 95% highest density credible interval for the expectations of the generalized linear model (GLM) on the sea urchin linear displacement of the mark-recapture experiment in the isoyake and vegetated habitat.

Click here for additional data file.

Supplemental Information 12 The result of the Bayesian GLM applied to the sea urchin group composition dataset.

The mean and 95% highest density credible interval for the expectations of the model on the sea urchin group composition of the mark-recapture experiment in the isoyake and vegetated habitat.

Click here for additional data file.

Supplemental Information 13 The raw monthly daily average wave height dataset.

The observed monthly daily average wave heights (m) and standard deviation. Data was recorded by a wave height logger installed at a point between the deep and shallow transects at the isoyake and vegetated habitat.

Click here for additional data file.

Supplemental Information 14 The raw monthly daily average temperature and light data.

The monthly daily average temperature (°C) and light (Photosynthetic Photon Flux Density, mol m−2 day−1) with their standard deviations. Data was recorded by a data logger installed on the deep and shallow transects of the isoyake and vegetated habitat.

Click here for additional data file.

Supplemental Information 15 The raw benthic quadrat data.

The monthly benthic quadrat from the deep and shallow transects of the isoyake and vegetated habitat. The benthic composition is expressed as the proportion of each benthic element (i.e., Coralline, Turf, Macroalgae, Substrate). The element “Other” (debris, benthic animals) was excluded from the analysis.

Click here for additional data file.

Supplemental Information 16 The raw sea urchin size-weight relationship data.

The model fit from the size and weight data from sampled sea urchins was used to predict the weight of the monthly sea urchin monitoring activity. The column “weight” is the predicted weight (grams) with its standard error from the column “se.fit”.

Click here for additional data file.

Supplemental Information 17 The raw monthly sea urchin monitoring data.

The monthly sea urchin monitoring data contains the urchin size, microhabitat (crevice, burrow, free-living), and size class (small, medium, large) for all species encountered (hcra: Heliocirdaris crassispina, dsav: Diadema savignyi, dset: Diadema setosum, tpil: Toxopnuestes pileolus, hpul: Hemicentrotus pulcherrimus). The species T. pileolus and H. pulcherrimus were excluded from the analysis due to scarcity.

Click here for additional data file.

Supplemental Information 18 The raw sea urchin mark-recapture experiment data.

The mark-recapture data contains information on the survey period (start, first recapture, second recapture), habitat (isoyake, vegetated), experimental trial (1–20), species (Heliocidaris crassispina, Diadema setosum), size of the tagged sea urchin, number of individuals together with the tagged sea urchin, linear displacement, and remarks (ok or lost). Lost sea urchins were excluded from the analysis.

Click here for additional data file.

Supplemental Information 19 The raw benthic rugosity data.

The benthic rugosity data was measured using the chain and tape method. The rugosity was calculated by dividing the straight-line chain length by the overlaid chain length.

Click here for additional data file.

Supplemental Information 20 The raw depth data.

The depth data was recorded using a depth logger deployed at the deep and shallow transects of the isoyake and vegetated habitat.

Click here for additional data file.

Supplemental Information 21 The raw wave height data.

The wave data was recorded using a wave height logger (Infinity—AWH-USB, Alec Electronics Co., Japan) deployed at the vegetated and isoyake habitat.

Click here for additional data file.

Supplemental Information 22 The raw temperature and light (PPFD) data.

The temperature and light (PPFD) data was recorded using a light/ temperature logger (HOBO Pendant MX2202, Onset Computer Corp.) deployed at the deep and shallow transects of the vegetated and isoyake habitat.

Click here for additional data file.

Supplemental Information 23 The average winter temperature data in Arikawa Bay from 2018–2021.

The average and standard deviation of the winter temperatures in Arikawa Bay from 2018–2021. The data was collected by instruments installed in an area close to the vegetated habitat.

Click here for additional data file.

Supplemental Information 24 The R script for analyzing the datasets used for the study.

This R script can be run using R studio and the Supplemental Datasets.

Click here for additional data file.

We thank the Arikawa Bay Fisheries Cooperative for providing resources and cooperation which allowed us to conduct the study.

Additional Information and Declarations

Competing Interests

Author Contributions

Data Availability

The authors declare that they have no competing interests.

Dominic Franco C. Belleza conceived and designed the experiments, performed the experiments, analyzed the data, prepared figures and/or tables, authored or reviewed drafts of the article, and approved the final draft.

Takeshi Urae conceived and designed the experiments, performed the experiments, analyzed the data, prepared figures and/or tables, authored or reviewed drafts of the article, and approved the final draft.

Shin-ichiro Tanimae conceived and designed the experiments, performed the experiments, analyzed the data, prepared figures and/or tables, authored or reviewed drafts of the article, and approved the final draft.

Kento Toyama conceived and designed the experiments, performed the experiments, authored or reviewed drafts of the article, and approved the final draft.

Akari Isoda conceived and designed the experiments, performed the experiments, authored or reviewed drafts of the article, and approved the final draft.

Gregory N. Nishihara conceived and designed the experiments, performed the experiments, analyzed the data, prepared figures and/or tables, authored or reviewed drafts of the article, and approved the final draft.

The following information was supplied regarding data availability:

The data used for the analyses and R code for analyzing the data are available in the Supplemental Files.

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
