# Peer review of "The behavior of sympatric sea urchin species across an ecosystem state gradient"

_PeerJ, doi:10.7717/peerj.15511_

## Round 0.1 · original submission · Major Revisions

I agree with the reviewers that the manuscript is, in general, well written although it needs some modifications which were indicated by the reviewers.

Reviewer 1 ·

Excellent Review

This review has been rated excellent by staff (in the top 15% of reviews)
EDITOR COMMENT
The comments made by the reviewer are very thorough and constructive. The reviewer also made a critical comment on the lack of connectivity between the title and introduction, with the actual results and discussion.

Basic reporting

Shifts from productive kelp beds to impoverished sea urchin barrens and the role of sea urchins in such shifts attract attention of the researchers for many years, therefore, the reviewed manuscript is of certain interest.

When reading Introduction, I believed that the authors intended to study the abundance and behavior of sympatric sea urchin species in macroalgal and barren habitats in order to reveal behavioral traits of sea urchins important for understanding their role in the deforestation of marine ecosystems. However, after reading Results and Discussion, I found that the authors dealt with not the case of “phase shift” from macroalgal to barren ecosystem due to sea urchin overgrazing but, most likely, with the case of ecosystem changes due to anthropogenic factor (i.e., construction of concrete peer in Arikawa Bay). I found no results confirming sea urchin grazing pressure and, moreover, I have doubts that isoyake (barren) habitat in Arikawa Bay has appeared due to this factor and that this habitat is real sea urchin barren. It could be seen from comparison of Fig. 1 and Fig. 8 and from Discussion (Lines 592-613) that the construction of the concrete jetty in the early 1970s could be the main cause of deforestation of the area studied.

It is not easy to understand from the MS title and Introduction what was the main purpose of the study? In Introduction, the authors reported “We monitored the abundance and distribution of urchin species along a vegetated and an isoyake (barren) area for over a year (September 2020 – December 2021) and also analyzed the movement patterns of the two most abundant sympatric urchin species, Diadema setosum and Heliocidaris crassispina through a mark-recapture experiment. We hypothesize that the substrate component and environmental conditions affect sea urchin movement patterns while the species and size determine microhabitat preference” (Lines 130-135). It is unclear from such formulation of the hypothesis tested what new contribution to the study of “the mechanism that drives urchin mediated phase shifts” (Line 101) did the authors intend to make? From Introduction (Lines 123-131), one can understand that sea urchins’ abundance and behavior would be studied in macroalgal and barrren habitats”, i.e., in two contrasting habitats. However, judging the title of the reviewed MS, the authors promised to study “The behavior of sympatric sea urchin species across an ecosystem state gradient”. I think that the study “across an ecosystem state gradient” should include not only two extreme states (macroalgal and barren) but also some transitional stages. Moreover, taking into consideration that the authors did not obtain any results on the role of sea urchin species in “phase shift” from macroalgal to isoyake/barren habitat in Arikawa Bay, they paid surprisingly much attention to sea urchin mediated phase shifts (Lines 86-109) as well as phase shifts in the state of the forest, lake and other ecosystems (Lines 56-65). I believe that, in terms of the results obtained, the authors should revise the title and Introduction and probably to formulate the study purposes and research problems more correctly.

Experimental design

Experimental design and methods are not described clearly.

1. (Lines 140-142) “The study site presents a case of contrasting ecosystem states where a vegetated and an isoyake habitat are spatially adjacent but physically separated by a concrete structure”. Are you sure that this “case of contrasting ecosystem states” has appeared due to grazing pressure of sea urchins and herbivorous fish (Lines 611-613) but not the construction of the concrete jetty in the early 1970s (Lines 601-602, Figs. 1 and 8)?

2. (Lines 148-149) “Within each ecosystem, a 20-meter stretch of rocky reef was selected for monitoring in deep (ca. 4 m) and shallow areas (ca. 2 m at IH and ca. 3 m at VH)”. Does it mean that the difference in depth between “deep” and “shallow” areas was 2 m and 1 m at IH and VH sites, respectively? Clarify please.

3. Seasonal macroalgal blooms (Lines 155-162) occurred during the spring to early summer at each site studied, both at vegetated habitat (VH) and isoyake habitat (IH). Fig. 5A demonstrates that macroalgae were present at both sites during most part of the study period. Are you sure the IH area matches the definition of "barren"?

4. (Lines 168-170) It should be reported how many photos were taken from each transect and from each deep and shallow area for each month.

5. (Lines 171-172) It should be reported how many such "2-meter-wide swath at 1 m intervals (2 x 1 m) along the transect" were studied per each habitat per month. If I understood correctly, sea urchins of each species were determined and counted under natural conditions? It should be clarified.

6. (Lines 173-176) “Urchin body width was measured using a caliper” – not width but diameter. Information about studies of sea urchin biomass is very scarce. It should be reported how many sea urchins of each species and with what periodicity (every month ?) were collected from each shallow and deep area at each site, and how they were weighed and measured. This information must be preceded by information that “Five urchin species occur at the study site…” (Line 176). How many size classes of sea urchins were distinguished and what is the range of these size classes?

7. (Lines 189-195) “Environmental conditions were recorded using data loggers.” It should be reported here whether data loggers were installed during the whole study period or not. At what depth they were installed at each site? Difference in depth between sites can influence data obtained and their interpretation. The authors report that they placed all data loggers between the deep and shallow transects at each site. However, Fig. 2 A, B demonstrates the data for light intensity and temperature both for shallow and deep areas. Please clarify.

8. Data analysis (Lines 227-339) was made using generalized additive model (GAM) or generalized linear model (GLM). This subsection contains many equations and description of predictors chosen for each model to explain differences in environmental conditions and biological parameters between the sites studied. However, in the Results, I could not find clear explanations regarding the most important (significant) predictors responsible for variations in sea urchin density and biomass, for example. The authors use the terms “high”, “low”, “less”, “greater” and do not indicate what differences were significant and what models were the best.

Validity of the findings

Given the uncertainty in the formulation of the goal and objectives of the study as well as in description of experimental design see above), it is very difficult to assess the validity of the findings. Based on the results obtained, I have doubts that isoyake habitat in Arikawa Bay can be considered as barren habitat which appeared due to sea urchins’ grazing activity. However, I believe also that the authors have done laborious large study and obtained a lot of results on the state of two sea urchin habitats, one of which can be considered as background (vegetative habitat, VI) while the other as impacted (isoyake habitat, IH). I suggest the authors to consider the possibility to focus not on sea urchin overgrazing and “phase shift” in marine ecosystem state but on ecosystem changes under anthropogenic influence. The manuscript should be revised.

Some other comments are given below and in attached PDF of the MS reviewed.

Additional comments

1. It should be indicated in “Introduction” that altogether, 3 sympatric sea urchin species were studied – Diadema setosum, D. savignyi and Heliocidaris crassispina. I also have to note that it is more correct to use the term “sea urchin” instead of simply “urchin”.

2. In “Introduction”, the authors paid much attention to the term "isoyake" (Lines 72-84). As is evident from literature, “isoyake” means the same that “barren”, and the authors also say about it (Lines 22, 72). What is the principle difference between “isoyake” and “barren” states of marine ecosystems? What is the principle difference in the causes for “phase shifts” from macroalgae forests to isoyake/ barren states? Graham, 2010 (cited in the MS reviewed) did not find important differences between “isoyake” and deforestation processes described for high- to low-latitude kelp systems worldwide. Fujita et al. 2013 and Ishikawa, Kurashima 2020 (cited in the MS reviewed) also used the term “barren” in their studies on the role of the sea urchin Diadema setosum in deforestation. The authors should clarify why they prefer to use "isoyake" instead of “barren”.

3. In the Results, there are references to 9 figures included in the MS body and 11 tables included in the supplemental material but not indicated as Table S1, Table S2 etc. All the tables have no titles. It should be corrected.

4. Excel Tables S1-S8 submitted in the supplemental material are not readable, unfortunately, probably because of Excel version incompatibility or using of Macro-enabled Excel. To help the readers, Excel Tables should be saved in Excel or Excel 97-2003 format. In addition, I found no references to Excel Tables S1-S8 as well as to Fig. S2-S8 (submitted in the supplemental material) in the text.

5. Description of algae communities at the sites VH and IH is not satisfactory (Lines 154-162, 363-369). Judging by Fig. 4A, macroalgae were present during much of the study period in both areas. The percent benthic cover of coralline algae in shallow area at VH site was higher during the whole study period than that in shallow area at IH site. It was somewhat higher also in deep area at VH. On what basis were the sites VH and IH determined?

6. Fig. 7. “Urchin densities that describe a discontinuous phase-shift”. How densities can describe anything? Incorrect title.

Annotated reviews are not available for download in order to protect the identity of reviewers who chose to remain anonymous.

·

Basic reporting

Please see additional comments below.

Experimental design

Please see additional comments below.

Validity of the findings

Please see additional comments below.

Additional comments

This paper compares environmental conditions and the densities ad behaviours of sympatric species of urchins between two spatially adjacent sites in Japan with different environmental and benthic conditions where the differences in benthic characteristics mimic two separate phases of kelp bed-urchin barren phase shifts. The authors assess environmental parameters between the two sites and observe differences in densities and movement and sheltering behaviour of two urchin species at each site. The authors provide inferences of why behaviour differs and what drives the environmental differences between the sites. Overall, I very much liked this manuscript. It was very well written, and I thoroughly enjoyed reading it. I thought the authors did a nice job designing the experiments and all data interpretations were reasonable. The data are useful, interesting, and certainly worthy of publication.

Nonetheless, I do have a few comments for the authors to consider. Notably, I thought that the Introduction placed a large emphasis on ecological phase shifts, but the Discussion didn’t really bring the results of the experiments back into the light of ecological phase shifts in a meaningful way. Here, I suggest either adding Discussion on ecological phase shifts (I provide an example on how to do this in the specific comments below; see my comment on Line 342), or focus the Introduction more on the impacts oof human activity on habitat disturbance and effects on localized communities. Additionally, I thought that some aspects o the methodology could benefit from increased detail and transparency, and I provide some suggestions for further considerations in the Discussion.

Based on the number of comments and their content, I would call this a moderate revision, though I want to be clear that I do think this paper is worth publishing. If the authors have any questions or concerns regarding any aspect of my review, they should feel free to contact me directly at jeffery.clements@dfo-mpo.gc.ca


Specific comments

1. Lines 25-26: The English grammar could be improved here. For example, this should read "The benthic rugosity at both sites was also surveyed." While the paper is quite well written throughout, there are some other examples of these instances throughout the text. I would suggest having a fluent English-speaking scientist familiar with the subject matter review the manuscript for English language and grammar if possible.

2. Line 31: The word "experiences" inn this sentence should read "experienced", so that the manuscript is always written in the past tense. An example of where a proficient English-speaking proofreaders would be valuable.

3. Lines 36-39. This is a very long sentence. I suggest breaking it into two sentences.

4. Lines 40-41: I think explicitly indicating the size is worthwhile here since the reader has no context for what a small, medium, or large urchin is. Also see my comments re. size classes in the Methods.

5. Line 43: The word "shows" should read "showed" as per my comment on past tense above. Please be sure to check the entire manuscript carefully for consistency.

6. Lines 56-57: This sentence is quite vague and incomplete. What anthropogenic activity. The impact of anthropogenic activity on what? Ecosystems do more than "lessen the impact of anthropogenic activity. I would consider removing this sentence and starting the paragraph with something like "The resilience of natural ecosystems to natural and anthropogenic stressors is limited and..."

7. Lines 88-90: And, under certain circumstances, they can even actively predate on animals that are widely considered to be predators of the urchins themselves (at least in the lab; a neat example of predator-prey role reversal!)

Clements et al. 2021. Ethology 127: 484-489.

8. Lines 102-103: It is not clear what specific "behaviour" the authors are referring to here. Feeding behaviour, perhaps? Some clarity is needed.

9. Line 125: Perhaps I am wrong here, but I think the authors mean to say that "...a particular species may display different behavioural patterns to other co-occurring species...". The term 'conspecific' means "off the same species", and I think the authors meant to convey that different species may behave differently. Please check and amend if necessary.

10. Lines 128-130: Cool! Sounds like a near system to work in!

11. Line 134: Again, "hypothesize" should read "hypothesized" for past tense. Please check the manuscript carefully throughout.

12. Lines 141-142: Some aerial or underwater images of the two habitats would be nice so that the reader can visualize the sites. If that authors have any photos, I suggest including them in Fig 1 with the map.

13. Lines 141-142: Some aerial or underwater images of the two habitats would be nice so that the reader can visualize the sites. If that authors have any photos, I suggest including them in Fig 1 with the map.

14. Line 187: Please provide some written rationale for why points labelled as "other" were excluded.

15. Lines 189-195: I'm a bit confused as to why a single logger was placed between the deep and shallow transects, rather than placing a logger at each of the transects. Please provide some rationale or explanation for this.

16. Line 200: Missing comma after "long-spined black urchin"

17. Lines 214-215: How many urchins were not recaptured? I.e., what was the recapture success rate?

18. Lines 232-236: I appreciate the detailed analysis the authors provide here - it is analytically strong - well done! One thing i would like to see is some rationale for the distribution choices. Why did you choose the particular distribution you did for each dependent variable? This is done for algae on Lines 237-240 below, but it is missing here.

19. Lines 254-256: Again, some rationale for the choice of priors here would be useful here and for the other subsequent analyses.

20. Lines 277-282: The size classes are never strictly defined in the text and really need to be, and no rationale for grouping sizes into these specific classes is provided. While this is done in Fig. 5, size ranges for each size class should be explicitly stated here in the Methods. Also, from the ranges given in Fig. 5, the cut-off between a small (1-3cm) and medium (4-5cm) is not clear; for example, what was an urchin with a width of 3.5cm classified as? It is also not clear why the size ranges of each size class are not standardized. For example, the small size class has a range of 1-3cm (2cm span), while the medium size class range is 4-5cm 9only a 1cm span), and the large size class has an infinite span (>5cm). It would be worthwhile showing the entire distribution of sizes from the urchins obtained so the reader can see the true variation in sizes for the urchins in this study. I would suggest doing this by species (does each species occupy the same total size range?). Finally, there needs to be some rationale for dividing the size classes as the authors do, and it needs to be recognized whether these classes were defined before or after data collection , else one runs the risk of categorizing sizes in a post hoc manner that results in bias when analyzing and interpreting the data.

21. Lines 290-310: the symbol "pi" is typically associated with a precise number (i.e., 3.14...). This may confuse some readers and I would suggest using a different symbol.

22. Line 342: Since environmental conditions differed between these two sites at the same time, it makes biological comparisons to other ecological phase shifts difficult, as these phase shifts occur in the same place at different times, which may or may not have similar environmental conditions. As such, it is hard to say whether differences in urchin behaviour observed here between the two habitats are driven by aspects of the benthic habitat itself (i.e., vegetation and rugosity, which would bee comparable to an ecological phase shift) or the different environmental conditions between the two sites (which likely differ due to the concrete barrier between the site, and which may or may not be comparable between phases of an ecological phase shift). The effect of these spatial differences in environmental conditions, their influence on the biotic parameters of urchins measured in this study, and their comparability/applicability to conditions at the same spatial location during temporally different ecological phases should be explicitly mentioned in the Discussion. This is particularly important given the focus on phase shifts in the Introduction.

23. Line 374: Why was "D. setosum expected to be the most common..."? I don't remember reading this as a specific hypothesis. Should this simply read "D. setosum was the most common...'

24. Lines 386-412: These results and their interpretation are precisely why more detail ad transparency are needed regarding the choice of size classes.

25. Line 416: Again, why was his "expected"? There was no direct, logical hypothesis presented for this to be expected.

26. Line 438: "evaluate" should be "evaluated". A reminder to check the manuscript thoroughly for consistent use of past tense.

27. Line 440: Again, need to use past tense: "...were highly adaptable and behaved differently..."

28. Line 518: Or is the barren state controlled by environmental conditions (sedimentation)?

29. Lines 529-531: Happy to see this limitation mentioned based on my comments on biomass estimations in the Methods. Good to acknowledge this - nicely done!

30. Lines 535-537: A good suggestion also due to the inherent variability of wet weight measurements which ca be an inaccurate representation of actual biomass. Good stuff!

31. Lines 558-560: Could increased movement in the IH also perhaps be due to it being a calmer, lower energy environment?

32. Lines 572-574: Perhaps the vegetation itself in the VH provided some refuge from predators, which could explain the more varied group sizes in VH.

33. Discussion, general comment: One thing I was left wondering about was whether the concrete barrier could act as a vector for speciation in this system. There were clear behavioural differences between the IH and VH and the barrier has been up for 50 years– do the authors think this could be actual adaptation (i.e., evolution), or is this just phenotypic plasticity? Is there any evidence of restricted gene flow between the two sites, either from genetic studies or hydrodynamic larval dispersal modeling? If gene flow is restricted ad there are some genetic differences, this is a cool example of human intervention driving an evolution on very small spatial scales.

34. Lines 622-644: The discussion is already quite long, but it is very well written (well done!). There was nothing new here in the conclusions that I didn't already read and gather from the Discussion. I am therefore not convinced that this conclusions section is necessary.

35. Figures, general comment: Very nice figures!

---

## Round 0.2 · Major Revisions

I agree with the reviewer's detailed comments, and that the abrupt changes in the measurement of environmental conditions between the two versions of the manuscript are alarming. Please kindly address all concerns raised with a point-to-point rebuttal.

Reviewer 1 ·

Basic reporting

I have carefully read the manuscript (MS) revised by the authors as well as answers to my comments. In the revised MS, the authors do not insist on their previous opinion regarding sea urchin overgrazing as the main cause of different state of vegetated and isoyake habitats (VH and IH) in Arikawa Bay, and this is good because their results do not support this point of view. The authors also took into consideration other comments and made corresponding changes in the text and illustrations. However, after reading the revised MS, I have additional comments which concern mainly the “Methods” section.

Experimental design

My main concern is regarding information about the measurement of environmental conditions at the study sites. In previous version the authors reported that “Environmental conditions were recorded using data loggers. For each habitat, one depth logger (HOBO U20, Onset Computer Corp., Bourne, Massachusetts, USA) and one wave height logger (Infinity-WH AWH-USB, Alec Electronics Co., Japan) was deployed between the deep and shallow transect. One temperature and light logger (HOBO Pendant MX2202, Onset Computer Corp., Bourne, Massachusetts, USA) was also placed between the deep and shallow transects. The instruments recorded environmental data at 10-minute intervals for 20 – 25 days before retrieval and data offloading” (Lines 189–195). The revised MS contains the following information: “Environmental conditions were recorded using data loggers. For each habitat, one wave height logger (Infinity-WH AWH-USB, Alec Electronics Co., Japan) was deployed at a point midway between the deep and shallow transects. For each deep and shallow belt transect, one depth logger (HOBO U20, Onset Computer Corp., Bourne, Massachusetts, USA) and one temperature/ light logger (HOBO Pendant MX2202, Onset Computer Corp., Bourne, Massachusetts, USA) was installed. The instruments recorded environmental data at 10-minute intervals for 20–25 days each month before retrieval and data offloading” (Lines 196–202).
What information is true? If the authors used the temperature/light loggers for each deep and shallow transects at each site, they must obtain tens of thousands of data for each transect. They could derive from these data hourly averaged, daily averaged etc. values of temperature and light intensity and had no need to use GAM/GLM analysis. In result of unclear description, I cannot understand why the data in Table S3 and Fig. 2 “recorded by instruments during the study” are named as “The average daily” whereas they seem to present monthly average values of temperature, wave heights and light levels (PPFD). There are 13-16 points of direct observations of the graphs (approximately each month). In addition, it is unclear from description of the methods how exactly the daily light level was determined. Did the authors take into consideration the whole day or light time only? Photon flux density should be equal to zero at night? Why minimum PPFD is > 0? (see Results, Line 357 and Fig. 2B).
There are some comments to subsection “Urchin mark-recapture experiment” (Lines 204-222). Due to unclear description in Methods, I cannot understand what is the difference between 0 and 24 h in Fig. 7 and Table S9? There were only 2 measurements per day (24 and 12 h), see Methods. If “0-hr” is the initial point (start), it should be noted both in Methods and illustrations. In this case, at what time of the day did you measure linear displacement and group size of sea urchins? And how did you measure linear displacement of sea urchins at point denoted as 0-hr (see Table S9, Fig. 7)? By what way?

Some comments to data analysis. I am not agree that a Bayesian approach for all analyses was a good idea. Probably, it is rather good for description of the “effects of time (i.e., survey months) and space (i.e., IH, VH, deep, shallow) on aspects such as urchin population, behavior and benthic characteristics”. However, if you have obtained the data of instrumental measurements of environmental variables at 10 min intervals, it much more informative to present them as direct observations instead of Bayesian distribution.
As to recommendation of the authors given in their rebuttal letter: “Furthermore, you can also check the histograms of the kernel density estimates of the model observations and model predictions in the supplementary information (Fig. S2 – Fig. S14). This can give the reader a qualitative idea of how well the model approximates the observations.” – I have to note that Fig. S2 – Fig. S14 are absent in the supplementary material presented by the authors (see also Line 349).

Validity of the findings

I made a number of comments to “Results” section (see PDF-file attached). The most important are the following:
1) The ranges of the light level at the IH and VH sites (Lines 357-358) seem to be not correct in the case if the light intensity was measured throughout the whole day. The minimum value must be zero. In addition, maximum light levels are very low at both sites and correspond to approximately 150–180 PAR. In our studies, for example, we have determined approximately 300–400 PAR at a depth of 7 m in sunny day. Sea water at both IH and VH sites seem to be very turbid. However, the authors did not discuss this fact in connection with algae productivity.
2) Presentation of the results obtained does not contain any information about the significance of differences. For example, are the “rugosities for the deep (1.16) and shallow (1.23) areas for IH, compared to the deep (1.38) and shallow (1.37) areas of VH” (Lines 367-368) significantly different between sites? From the all description of the occurrence rates of sea urchin species at different sites, it is not clear what GAM model and what predictors were the best to describe habitat preference (see Lines 292-294: The predictor variables were location (2 factor levels: IH, VH), transect (2 factor levels: deep, shallow), species (3 factor levels: D. savigny, D. setosum, H. crassispina), microhabitat (3 factor levels: pit, crevice, free-living)... “. There is no information on significant differences in sea urchin density and biomass between species and sites/transects as well.
3) Description of the movement patterns of H. crassispina and D. setosum (Lines 422-443) needs to be corrected according to the comments made above. Authors should indicate what the point 0-hr does mean and at what times of the day the movement measurements were made. It should be also indicated how many repetitions of this experiment were made. In addition, Table S9 (Line 425) is a copy of Table 2 (Line 438). The legend to Figure 7 should be also corrected.

Additional comments

1. In Discussion, I have to note some incorrect citations. Lines 553-554: Tuya, Martin and Luque, 2004 did not conduct “similar study” (similar to the present study reviewed, I believe). The following claim “Indeed, hunger has been known to override urchin predator avoidance behaviors (Belleza et al., 2021)” (Lines 559-560) is incorrect because (1) the authors did not study sea urchin responses to predation threat in the present work and (2) Belleza et al., 2021 studied only one sea urchin species, H. crassispina. Meanvile, there are examples of different behavior of cohabitant sea urchin species to predator threat in the presence of food (Zhadan and Vaschenko, 2019, PeerJ 7:e8087 DOI 10.7717/peerj.8087). Well-fed sea urchins Mesocentrotus nudus and Strongylocentrotus intermedius exhibit different responses to damaged conspecifics: M. nudus leaves the food source for a long time while S. intermedius forms dense groups near the food source. Therefore, the authors should not spread the results of their study (Belleza et al., 2021) to all sea urchin species.

2. References in the Reference list are formatted not according the PeerJ journal reference format: List of authors (with initials). Publication year. Full article title. Full title of the Journal, volume: page extents. DOI (if available). In addidion, the authors should check the citation because not all References are present in the Reference list.

Some other comments are given below and in attached PDF of the MS reviewed.

Annotated reviews are not available for download in order to protect the identity of reviewers who chose to remain anonymous.

---

## Round 0.3 · Minor Revisions

The manuscript is almost ready for acceptance, pending another round of minor revision. I look forward to reading the revised version of the manuscript.

Reviewer 1 ·

Basic reporting

I have read the manuscript (MS) after second revision and the answers of the authors to my comments and found that the authors have significantly improved the text and illustrations. They also have given satisfactory answers to my questions regarding instrumental methods applied. I made several small comments (see PDF-file attached) which can help improve the perception of the article by readers.

Experimental design

In the present version of MS, methods are described with sufficient details.

Validity of the findings

In the present version of MS, the results obtained are described clearly enough

Additional comments

I think that the article should be accepted after small revision. I think that I would NOT necessarily need to re-review MS.

Annotated reviews are not available for download in order to protect the identity of reviewers who chose to remain anonymous.

·

Basic reporting

See "Additional comments" below for my comprehensive set of comments.

Experimental design

See "Additional comments" below for my comprehensive set of comments.

Validity of the findings

See "Additional comments" below for my comprehensive set of comments.

Additional comments

The authors have largely addressed the reviewer’s comments, as far as I can tell. I agree with the authors’ responses to most comments, including the need for “significant” differences (explanatory power suffices for this analytical approach, and I commend the authors for using a Bayesian approach rather than conforming to NHSTs.

However, I tend to agree with the reviewer that the authors likely don’t need to fit GLMs/GAMs to environmental data given that they have raw measurements at high frequency. Herein, the reviewer is not saying that the approach is “excessive”, but they are saying that these analyses are not as precise as providing the actual, high frequency measurements obtained – and I agree. I do, however, see the usefulness of the model fit for the long chunk of missing light data for the shallow vegetated habitat. But I do tend to agree with the reviewer that including the raw measured values for other parameters are more valuable/applicable than monthly averages fit with complex models, as the measured parameters are what the animals actually experienced during the experiment. I would suggest at least including lines in Figure 2 connecting the points, and perhaps adding a supplementary figure displaying raw daily averages (not monthly daily averages). I also suggest including ALL of the raw data for environmental data in supplementary files (i.e., all point measurements) alongside the monthly daily averages, and linking to that in the text, so readers can access all of the raw data.

A small suggestion is to rename the file “Supplementary Information” to “Supplementary Figures S1-S14” to aid readers in easily locating supplementary figures.

Finally, re. low light levels: perhaps it is worth speculating in the Discussion about turbidity/algal productivity in explaining why light levels were low, but being explicit that you did not directly measure this.

---

## Round 0.4 · Minor Revisions

I agree with the authors' responses to the reviewer's comments. However, I noticed there are some grammatical errors and language inconsistencies that needs proofreading before acceptance. Kindly proofread the whole manuscript again. Some (but not all) are listed below:
1. Line 20: due to their 'grazing ability'?
2. Line 21: the habitat
3. Line 26-27: normally when we use 'most', it will only refer to one species and not two. Suggest to change to 'was conducted on the TWO most abundant sea urchins'
4. Line 70: do you mean 'is A Japanese concept'
5. Line 101: this sentence seems hanging. please revise.
6. Line 212: do you mean each individual was separated by 5m during collection?
7. Line 299: 'were' analyzed?
8. Line 416: 'In' should be 'in', not capitalized
9. Line 476-478: this sentence needs to be rephrased.

---

## Round 0.5 · accepted · Accept

I thank the authors for patiently following through so many rounds of reviews. I am looking forward to reading the published version of this manuscript!